# A scaling law to model the effectiveness of identification techniques

Luc Rocher [1,2,3] ✉, Julien M. Hendrickx[2,5] & Yves-Alexandre de Montjoye[3,4,5] ✉

AI techniques are increasingly being used to identify individuals both offline and online. However, quantifying their effectiveness at scale and, by extension, the risks they pose remains a significant challenge. Here, we propose a two-parameter Bayesian model for exact matching techniques and derive an analytical expression for correctness ($\kappa$), the fraction of people accurately identified in a population. We then generalize the model to forecast how $\kappa$ scales from small-scale experiments to the real world, for exact, sparse, and machine learning-based robust identification techniques. Despite having only two degrees of freedom, our method closely fits 476 correctness curves and strongly outperforms curve-fitting methods and entropy-based rules of thumb. Our work provides a principled framework for forecasting the privacy risks posed by identification techniques, while also supporting independent accountability efforts for AI-based biometric systems.

Anonymity is an essential property of democratic life that underpins freedom of expression and digital rights. Anonymity appears *naturally*, in the absence of identification, surveillance, or traceability in mass societies[1]. Natural expectations of anonymity have been recognized by regulators and legal scholars, and anchored by legal tests[2]. Anonymity can also be defined *normatively* as a process that "obliterates the link between data and a specific person" according to Barocas and Nissenbaum[3]. In this case, anonymity appears by design: it is e.g. built into data processing systems such as the private network Tor[4] and COVID-19 contact-tracing apps[5–7], or results from de-identification methods[8].

Advances in computational power and machine learning are, however, progressively challenging both natural and normative conceptions of anonymity. Identification techniques—matching individuals across digital traces—can be broadly divided into three categories: 'exact', 'sparse', and 'robust' matching. Exact matching dates back to 1956[9] and was made famous by Latanya Sweeney's re-identification of the Governor of Massachusetts's medical data in 1997[10] from his ZIP code, gender, and date of birth. Exact matching refers to the process of identifying a known individual in an anonymized dataset using known quasi-identifiers, e.g., a few pieces of demographics[10–12]. Exact matching has since been applied broadly to identify online users from, e.g., web browser fingerprints[13]) and cryptocurrency transactions[14]). Sparse matching was then proposed ten years ago to extend exact matching to sparse 'set-valued' datasets, with each record including a large set of points, e.g. all the goods purchased or locations visited by an individual. Research on 'unicity' in 2013 showed that a few points are enough to identify individuals in set-valued data[15]. Sparse matching has since been used to identify users from credit card transactions[16], mobile phone apps[17,18], or web browsing history[19]. Finally, recent advances in deep learning have accelerated the development of 'robust' matching techniques. Robust matching includes methods that identify an individual from noisy or approximate information, e.g., by learning similarity metrics from geolocation data[20,21] or human faces[22]. It also includes profiling techniques, e.g., capable of handling distributional shifts in datasets across time[23] and allowing them to learn how people write[24], communicate[25], or speak[26], in order to identify them.

Measuring the performance of an identification technique in the real world is, however, surprisingly difficult. While identification techniques are often tested in small-scale benchmarks, identifiability is known to decrease monotonically with the 'gallery size', the number of

[1]Oxford Internet Institute, University of Oxford, Oxford, UK. [2]Information and Communication Technologies, Electronics and Applied Mathematics (ICTEAM), Université catholique de Louvain, Louvain-la-Neuve, Belgium. [3]Data Science Institute, Imperial College London, London, UK. [4]Department of Computing, Imperial College London, London, UK. [5]These authors contributed equally: Julien M. Hendrickx, Yves-Alexandre de Montjoye. ✉e-mail: luc.rocher@oii.ox.ac.uk; demontjoye@imperial.ac.uk

individuals against which a target is compared to, as erroneous matches become more prevalent. Intuitively, it is e.g. much easier to identify someone amongst pictures of five people than pictures of a million people. The speed and shape with which erroneous matches appear depend on a number of factors, related to the identification technique and the underlying data on which it is trained and evaluated on, so far hard to predict. Some identification techniques, e.g., those relying on geolocation traces or images, have been shown to scale to large populations[27,28] while others, e.g., relying on writing style or smartphone app usage[17,18,29], have not.

We propose a Bayesian model to estimate how the performance of a specific identification technique, tested on a set of auxiliary information known by an attacker, scales with the gallery size ($n$). We focus first on exact matching with discrete tabular data, where the gallery can be partitioned into anonymity sets[30] of records with same quasi-identifiers. This partitioning encodes all the information required to calculate the correctness $\kappa(n)$, the fraction of individuals accurately matched in the gallery (e.g., if four records share the same demographics, they would form one set where each has 25% probability of being correctly identified using those demographics). We model anonymity set sizes as Pitman-Yor processes and derive an analytical expression for $\kappa(n)$ using two parameters, the entropy ($h$) and tail complexity ($\gamma$). We validate this model on exact matching identification with 400 galleries, selected by sampling sets of quasi-identifiers from real-world datasets (census, survey, web fingerprints) and synthetic datasets (geometric, Poisson, and Zipf distributions). For each gallery, we calculate frequency data of anonymity sets and estimate the maximum a posteriori (MAP) values for $h$ and $\gamma$. From $h$, $\gamma$, and $n$, we estimate $\kappa(n)$ with the model reaching a low RMSE error of 1.7 percentage points (p.p.).

From our general model, we then derive a functional form for $\kappa(n)$, which can be used as a general scaling law to forecast the correctness of exact, sparse, and robust matching. For instance, knowing that $\kappa = 0.99$ for $n = 100$ and $\kappa = 0.80$ for $n = 1k$, this allows us to 'extrapolate the curve' and forecast that $\kappa \approx 0.21$ for $n = 10k$. We validate this scaling law on 476 correctness curves from exact, sparse, and robust matching, achieving a low RMSE of 5.1 p.p. when forecasting the correctness in galleries ten times larger. We compare this approach to ad-hoc curve-fitting methods and rules of thumb, e.g. fitting $\kappa(n)$ to a polynomial function, an exponential decay, or a log-linear model, and show that these methods all result in large over- or under-estimations.

Accurately estimating if an identification that was successful in a small gallery will have low, moderate, or high accuracy in a much larger gallery is key to better evaluating these technologies. Firstly, with only access to a small-scale sample, researchers can now predict how many people would likely be re-identified in a large gallery without having to collect further data. Secondly, researchers can now use the scaling law to compare the reported performance of different identification techniques, often each validated on their own unique datasets with potentially different gallery sizes. Finally, they can also use it to compare different sets of auxiliary information (e.g. demographic information with admission dates to an hospital versus a noisy version with approximate dates) on the same probabilistic matching technique, to compare their applicability.

We see our method as a new tool to evaluate how successful an identification attempt can be in practice, helping not only measure the risk of re-identification in data release but also support evaluations of robust behavioral identification being deployed in high-risk settings[31] such as in hospitals[32,33], humanitarian aid delivery[34], or border control[32,35]. The scaling law can help test if identification techniques in high-risk settings are accurate enough at scale and in non-adversarial settings for the conditions in which they will be deployed.

## Results
We consider a study evaluating the performance of an identification technique. The study measures a standard privacy and biometrics task

called 'closed-set identification' (also called 1:N recognition or matching), of which we adopt the terminology. In this task, an adversary aims to match an unidentified 'probe' (a target individual's record) against a gallery of potential candidates, by comparing extracted features called 'auxiliary information' in the privacy literature (or 'feature vector' in biometrics). The study reports a metric termed correctness ($\kappa$), representing the average success rate of matching[36]. This metric is calculated by iteratively selecting each record in the gallery, assuming it to be the probe, and testing if the identification succeeds. Our objective is to forecast the correctness of this identification technique on a larger population of $n'$ records.

To define $\kappa$ formally, we initially restrict ourselves to exact matching attacks. We denote by $G = \{x_l\}_{l=1}^n$ the gallery of $n$ enrolled records. Each record $x \in G$ is a vector, from which auxiliary information $\phi(x)$ can be extracted. Each record $x$ belongs to an *anonymity set*[37–39] $S(x) = \{y \in G | \phi(y) = \phi(x)\}$, an equivalence class of records sharing the same auxiliary information. The larger the anonymity set, the harder is it to correctly match records in that set. The correctness $\kappa$ measures the probability that a probe $x$ matched using auxiliary information $\phi(x)$ is assigned its correct identity in its anonymity set[36]: $\kappa = \frac{1}{n} \sum_{l=1}^n |S(x_l)|^{-1}$. For instance, in a gallery of three records sharing the same auxiliary information (a single anonymity set of size three), each would have one chance out of three ($\kappa = 1/3$) to be correctly identified based on $\phi(\cdot)$. Conversely, if each record has its own unique auxiliary information, each would always be correctly identified ($\kappa = 1$).

### Modeling auxiliary information and identifiability
Our model assumes that each record $x \in G$ is drawn i.i.d. according to a distribution $X$ and $\phi(x) \in \mathbb{N}$ is an integer encoding the anonymity set of $x$. The expected correctness $\kappa$ is entirely determined by the distribution of anonymity set sizes. We denote by $\pi = \{\pi_i\}_{i \geq 1}$ the family of random frequencies of anonymity sets, with $\pi_i \geq 0$ and $\sum_{i \geq 1} \pi_i = 1$. Here, $\pi_{\phi(x)}$ represents the probability to draw the anonymity set of the record $x$. Formally, $\kappa$ can be expressed as:

$$\kappa(n) = \sum_{i=1}^{\infty} \pi_i \left[ \frac{1 - (1 - \pi_i)^n}{n \pi_i} \right] \tag{1}$$

We propose a general Bayesian approach based on Pitman-Yor processes to model the random frequencies $\pi$ and the correctness $\kappa$. The Pitman-Yor process, a generalization of the Dirichlet process, is a flexible *two-parameter* distribution over discrete distributions. Pitman-Yor processes have been used to study how online social networks grow[40], count object frequencies for image segmentation[41], build species sampling models in ecology[42], and can model a wide range of discrete distributions including heavy-tail discrete distributions. A Pitman-Yor process induces an 'exchangeable partition'[43] of arbitrary size. This is particularly useful here to model the frequencies of anonymity sets, since the ordering of anonymity sets does not matter. Pitman-Yor processes are governed by two parameters, a discount parameter $d \in [0, 1]$ and a concentration parameter $\alpha \in [-d, +\infty]$.

We derive the Pitman-Yor Correctness (PYC) model by setting a Pitman-Yor prior for the frequencies of anonymity sets $\pi \sim PY(d, \alpha)$. We reparametrize the prior to introduce the information content of the distribution $\pi$ to model independently $h$, the expected Shannon entropy, and $\gamma$, the tail complexity (see eq. (2)):

$$\begin{cases} h &= \mathbb{E}\left[H(\pi) | d, \alpha\right] = \psi_0(\alpha + 1) - \psi_0(1 - d) \\ \gamma &= \frac{\psi_0(1) - \psi_0(1 - d)}{\psi_0(\alpha + 1) - \psi_0(1 - d)} \end{cases} \tag{2}$$

with $\psi_0$ the digamma function[44] (SI Section S2.2). While $h$ measures the average uncertainty over all possible auxiliary information known by an adversary, $\gamma$ measures the heaviness of the tail (the higher $\gamma$ is, the heavier the tail of anonymity set sizes). See Discussion for an analysis of the impact of $h$ and $\gamma$ on the correctness.

We derive an analytical expression for the expected correctness using only $h$, $\gamma$, and the population size $n$ by integrating on the infinite-dimensional simplex from eq. (1) (proof in SI Section S2.2):

$$
\mathbb{E}\left[\kappa(n)\,|\,h,\gamma\right] \\
= \frac{v-1}{n(u-1)}\left[\frac{\Gamma(n-u+v)\,\Gamma(v-1)}{\Gamma(n-1+v)\,\Gamma(v-u)}-1\right] \quad \text{with} \quad \begin{cases} u = \psi_0^{-1}(\psi_0(1)-h\gamma) \\ v = \psi_0^{-1}(\psi_0(1)+h-h\gamma) \end{cases} \tag{3}
$$

with $\Gamma$ the standard gamma (factorial) function[44].

Importantly, our approach does not assume a particular parametric form for the frequencies $\pi$ of anonymity sets, nor prior knowledge on the support size[45].

## Model specification of PYC on empirical frequencies

We first validate the specification of the PYC model, by testing that (a) the Pitman-Yor process fits well typical frequencies of anonymity sets and (b) the PYC accurately predicts the correctness when fitted on empirical frequency data. Building upon previous work in statistics[45,46], we propose a fast and scalable method to obtain the maximum a-posteriori (MAP) estimates of $h$ and $\gamma$ from the empirical frequency distribution of anonymity sets (SI Section S2.3). These estimates can then be used to evaluate the equation (3).

We collect five corpora from publicly available sources (detailed description in SI Section S1.1): population census (USA[47], ADULT[48]), survey (HDV[49], MIDUS[50]), and web browser fingerprinting (WEB[13,51]). We create 250 datasets by selecting 50 subsets of attributes (columns) uniformly from every corpus, each with one unique set of auxiliary information $\phi(\cdot)$ which would be used by an attacker to perform an exact matching attack. We also create 150 synthetic data collections from Geometric (GEOM), Poisson (POISSON), and Zipf's law distributions (ZIPF). These distributions–traditionally used to model human data–capture a larger range of information content ($h$) and heaviness ($\gamma$). This allows us to validate suitability of the model and effectiveness of the fitting procedure across a broader range of scenarios, complementary to the real data collections.

The PYC model accurately estimates the correctness across data collections and population sizes, both for empirical data and synthetic data (Fig. 1a). Over all 400 studied datasets, we obtain a low bias of + 1.3 percentage points (p.p.) and a high accuracy with an RMSE of 1.7 p.p. (see Table S1). We hypothesize this small bias to be due to the choice of an informative PYP prior, in line with the literature[45]. Such informative prior could indeed weight less on strongly heavy-tailed distributions, leading in this case to a small positive bias for synthetic data and a small negative bias for empirical data.

This accurate prediction of $\kappa$ stems from a flexible model specification of the population. In Fig. 1b–i, we compare the empirical fit between the empirical frequency distribution and expected frequencies from the $(h^*, \gamma^*)$ MAP estimates for 8 specific datasets (5 real in orange and 3 synthetic in blue) each coming from a different corpus. Across datasets, they exhibit a good approximation of the empirical distribution, even for widely different forms of the random frequencies $\pi$ such as exponential (Fig. 1g) and heavy (Fig. 1i) tails. The same is true across corpora with an average Kullback–Leibler (KL) divergence[52] ($D_{\mathrm{KL}}(\hat{\pi}\,\|\,\pi)$) of 4.14 ± 7.55 bits (see Table S2).

Finally, Fig. 1b–i (inset) show how–once fitted to the empirical frequency data–the PYC model is able to correctly infer the correctness $\kappa$ across a wide range of subsampled population sizes from 1 to $n$. The eight panels also show that the correctness curve can take a wide range of decaying shapes, that our model captures well in each case. This accurate inference across population sizes suggests that the PYC model should be well suited to forecast $\kappa$ at larger population size $n'\gg n$ (see next section).

## Evaluating the success of exact, sparse, and robust matching from measurement data

The formal PYC model of identifiability is theoretically limited to exact matching, assuming that each individual is represented by a unique auxiliary information $\phi(x)$ that never changes over time. This assumption would be violated in sparse matching attacks, where an adversary can, e.g., aim to identify Alice in an anonymous mobility dataset from any set of $p = 4$ random locations she posted online. The correctness of sparse matching measures an average accuracy of identification across all potential sets of $p$ points known by an attacker (all equally likely to be selected in this threat model). In our example, some locations are more frequent than others and some sets of points $p$ are more frequent than others. The joint distribution of $p$ points is thus not equivalent when selecting different sets of $p$ points. Similarly, the assumption would be violated in a robust matching attack where an adversary uses a machine learning model to identify Alice in a medical imaging dataset, with a different X-ray taken days or months later.

We however believe that, despite the lack of theoretical guarantees, the PYC model captures the correctness of sparse and robust matching techniques, even if the auxiliary data is not necessarily fixed and discrete. The PYC model associates each individual in the population with a unique set of auxiliary information ($\phi(x)$). The main difference between exact and sparse matching techniques lies in the auxiliary information, as an individual can be identified by multiple sets $\phi(x)$ for the latter. Similarly, robust matching methods can also identify an individual from different sets of imprecise auxiliary information, by forming a robust representation or embedding of their behaviors. For sparse and robust matching, we can intuitively see $\phi(x)$ as a compressed latent representation of changing auxiliary information, akin to a unique fingerprint. In that case, the correctness is equivalent to the rank-1 identification rate of the matching performed by the attacker (SI Section S2.2).

In this article, our goal is to evaluate the correctness of an identification technique on a population $n'$ from access to only $t\geq2$ observations of the correctness $\kappa$ at population sizes $n^{(1)}, n^{(2)}, \ldots, n^{(t)} < n'$. To do so, we select the parameters $\hat{h}$ and $\hat{\gamma}$ minimizing the expected quadratic loss:

$$
R(h,\gamma) = \sum_{i=1}^{t} \log(n_i)\left[\hat{\kappa}(n^{(i)}) - \mathbb{E}[\kappa(n^{(i)})|h,\gamma]\right]^2 \tag{4}
$$

using a derivative-free method (Nelder–Mead[53]) and a logarithmic weighting. The latter aims to set more weight on larger sample sizes, associated with more accurate correctness measurements. Given $\hat{h}$ and $\hat{\gamma}$, the PYC yields an estimator $\mathbb{E}[\kappa(n')|\hat{h},\hat{\gamma}]$ for any population size $n'$. We call this method PYC-MB for 'measurement-based'.

We now evaluate the accuracy of PYC-MB on exact, sparse, and robust matching. For exact matching, we use the same discrete data corpora studied so far. We additionally use three corpora of unstructured set-valued data for sparse matching (installed apps on Android phones (APPS[54]), shopping carts (SHOPS[55]), and mobile call metadata (CALLS[27]); see SI section S1.2) and four corpora of machine learning experiments for robust matching (identification from face photographs (FACEREC[56]), GPS mobility traces (GEO[21]), interaction graphs with geometric deep learning (IIG[25]), written texts (TEXT[29]); see SI section S1.3). We selected recent identification techniques and auxiliary information, for which we either had access to the data (sparse) or precise measurements from peer-reviewed publications (robust). We did not re-implement robust identification techniques but used reported correctness, thus maximizing the applicability of our findings below.

We evaluate the accuracy of PYC-MB using the same task across identification techniques. For each identification, we compare the empirical correctness $\kappa(n')$ with the predicted correctness $\mathbb{E}[\kappa(n')|\kappa(n^{(1)}), \ldots, \kappa(n^{(t)})]$. We predict the correctness from a small

sample of $t = 50$ correctness measurement points, evenly spaced in log space from $n^{(0)} = 1$ to $n^{(t)}$ (see SI section S3.2). For robust matching, we rely on reported correctness scores from existing peer-reviewed publications. Therefore, some experiments have fewer than $t = 50$ data points and are not necessarily evenly spaced in log space. We test three sampling fractions $\mu = n^{(t)}/n'$ at 1%, 5%, and 10%.

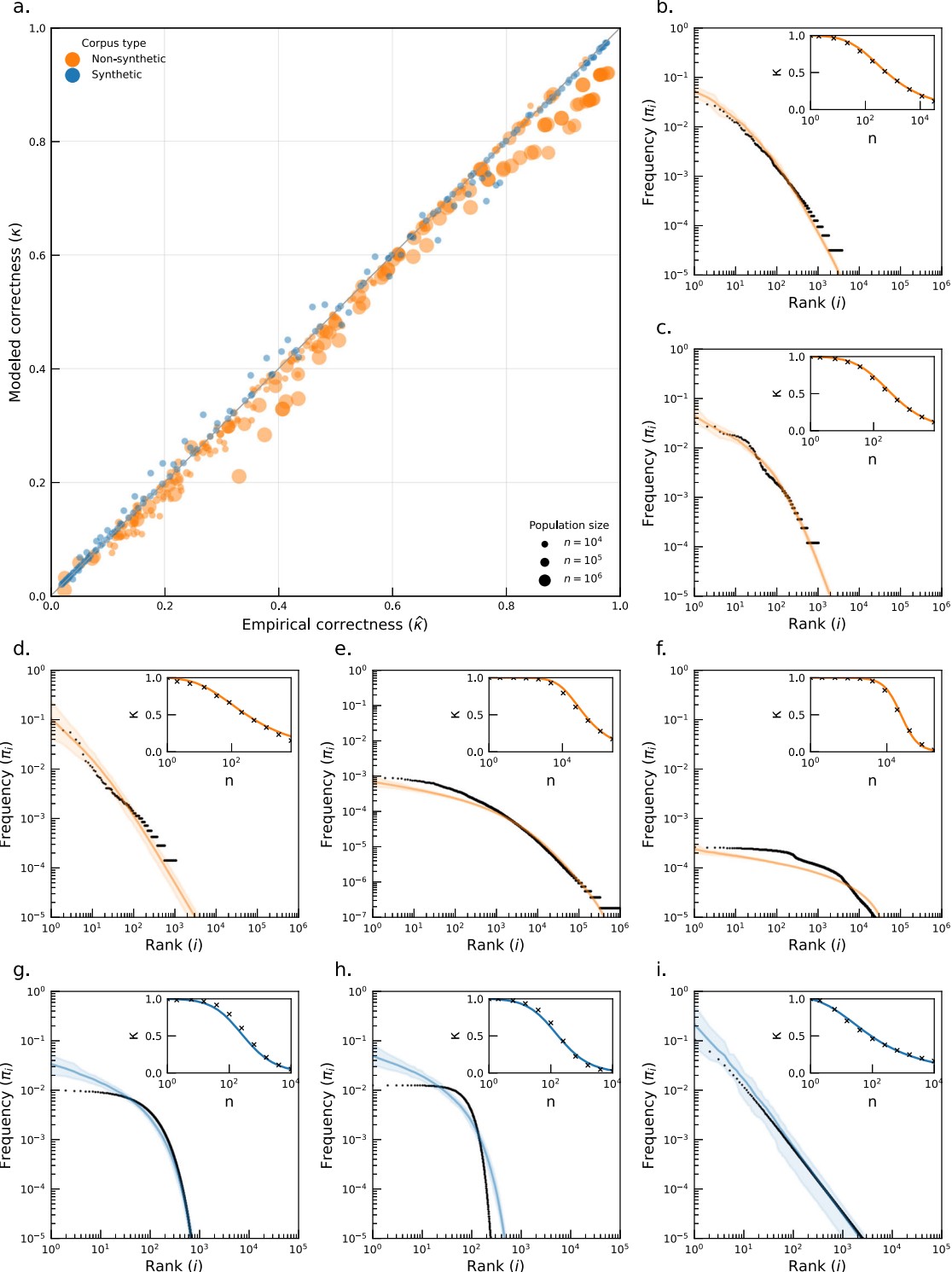

**Fig. 1 | Pitman-Yor processes model a wide range of discrete count distributions and the PYC correctly predicts their correctness. a** Empirical vs. estimated correctness for all 400 surveyed datasets, across all corpora. **b**–**i** Posterior predictive checks for the empirical frequencies $X_\Phi$. We report the rank-size distribution of empirical samples (black) and 95% CI sampled from the MAP estimate of the PYP (orange for empirical data, blue for synthetic data). We sample random frequencies of anonymity sets from $\pi \sim PY(h^*, \gamma^*)$, using stick-breaking representations, to obtain 95% confidence intervals on the inferred probability mass functions. (Inset) Empirical (black dots) and expected correctness according to the PYC model (solid line) for a population size ranging from 1 to $n$ individuals. **b** Demographics from the ADULT corpus (ADULT-1). **c** Demographics from the HDV corpus (HDV-1). **d** Demographics from the MIDUS corpus (MIDUS-1). **e** Browser fingerprints from the WEB corpus (WEB-1). **f** Demographics from the USA corpus (USA-1). **g** Synthetic Geometric corpus (GEOM-1). **h** Synthetic Poisson corpus (POISSON-1). **i** Synthetic Zipf corpus (ZIPF-1).

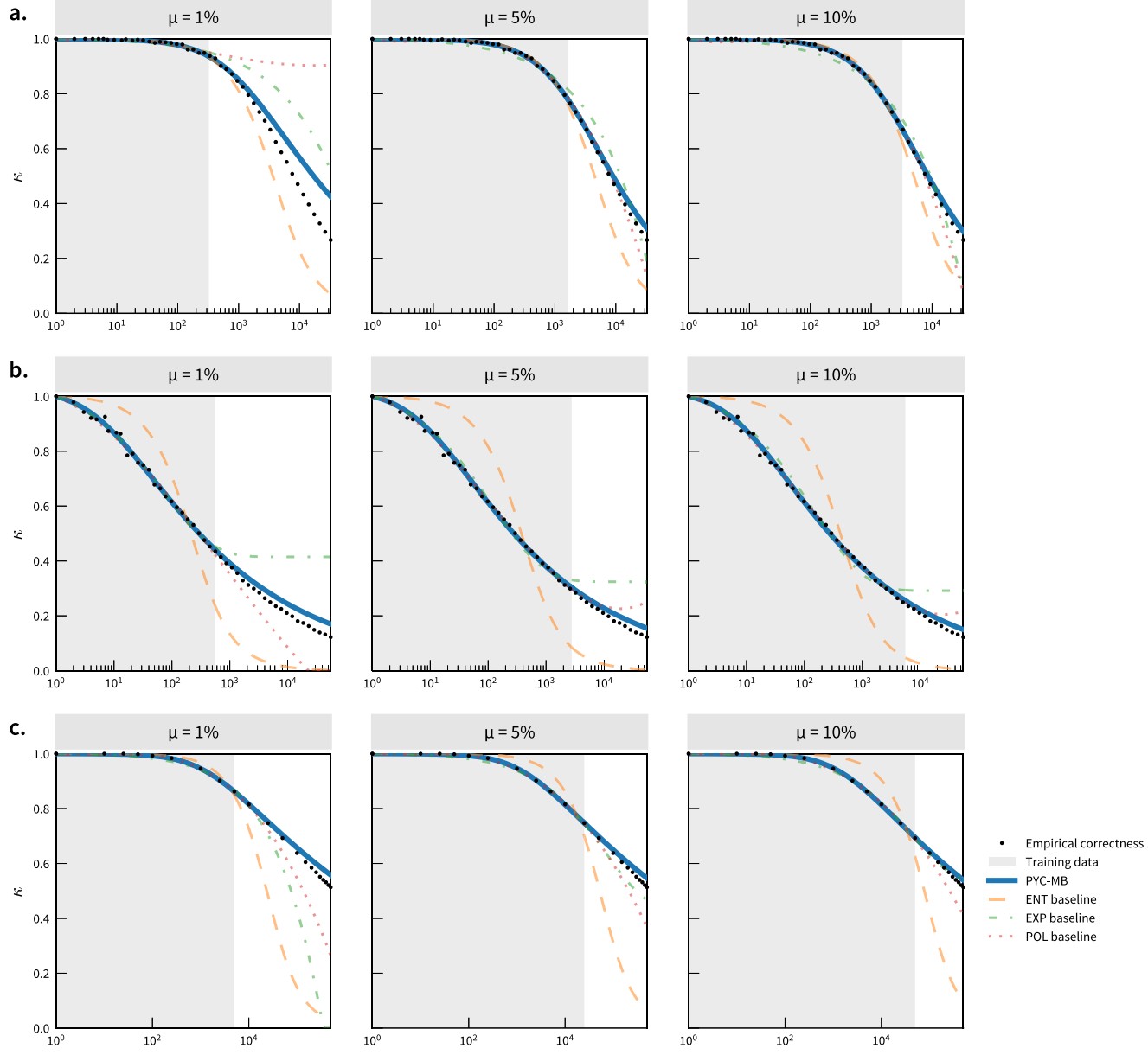

**Fig. 2 | The PYC-MB extrapolation method captures the correctness more accurately than previously-used heuristics and rules of thumb.** We perform measurement-based extrapolation of (**a**) exact, (**b**) sparse, and (**c**) robust matching attacks. We report the performance of the PYC-MB method compared to three other functional forms (ENT−Entropy baseline with no tail complexity, orange line; EXP−exponential decay function, green line; POL−polynomial function, red line; see Supplementary Note S3.2). We report the performance when trained on (**a**) exact matching (ADULT-1, using discrete demographics), (**b**) sparse matching (APPS-1, using 2 installed Android apps), (**c**) robust matching (GEO-1, ML-based mobile phone geolocation matching). We measure the empirical correctness up to $\mu \in \{1\%, 5\%, 10\%\}$ of the original data and, for each sampling fraction $\mu$, fit the four functional forms. We display the fitted correctness with solid color lines and the training part with a gray background. We display the empirical correctness with black dots. In all examples, the PYC-MB achieves high accuracy with good model specification. Figs. S1 to S17 include additional examples on all studied corpora and Tables S3 to S7 report RSME values for all samples and corpora.

Figure 2 shows that the PYC-MB model predicts well the correctness of (a) exact, (b) sparse, and (c) robust attacks from 1, 5, and 10% measurement samples, strongly outperforming the three other baselines we considered (see Discussion). PYC-MB obtains a low RMSE across corpora and sample size, reaching for instance a low RMSE of 5.1 p.p. when predicting the population correctness from a 10% exact matching study (Table 1). These accurate estimates stem from a good model specification, since the PYC-MB reaches an RMSE of only 1.3 p.p. when fitted on the complete population ($\mu = 100\%$ with $n' = n^{(t)}$). Table 1 further shows that PYC-MB obtains low RMSE across corpora and sample size, reaching, for instance, an RMSE of 3.6 p.p. (sparse)

and 10.2 p.p. (robust) when predicting the population correctness from a 10% sample.

Finally, we show how our method can be used in practice to reason about the scalability of four popular identification techniques from measurement data up to the world's population. We consider the identification of mobile phones from 1-hop interaction graphs[25] such as those collected by some contact tracing apps (IIG-1); authors from texts they wrote, a problem known as authorship attribution[29] (TEXT-1); human faces from photographs[56], a problem known as facial recognition (FACEREC-2); the identification of a browser through simple technical identifiers, a problem called browser fingerprinting[57] (WEB-2).

Figure 3 reports the scalability of identification for these four attacks up to 7.53B people using the PYC-MB model. While machine learning methods such as 1-hop interaction graph matching and authorship attribution can reach a very high correctness for small-scale populations, their effectiveness decreases fast as soon as the population of interest is above 100 to 1000 people. Indeed, both of them exhibit a near-geometric tail (low $\gamma$, 0.43 for IIG-1 and 0.33 for TEXT-1) with a fast transition from high to low correctness as the population size increases. This means that while, e.g., identifying the author of a

comment amongst a small set of known forum authors is a risk, trying to identify them within all Internet users of a country might currently work only for a few outliers with particular writing styles or topics.

Other methods, however, exhibit a very different scaling behavior. The correctness of the facial recognition technique we consider here (Google FaceNet V8, FACEREC-2) is roughly similar to both 1-hop interaction graph matching and authorship attribution for a population of 100 people. Yet, because of its high $\gamma$ of 1.0, the correctness of the facial recognition technique only decreases very slowly with population size. Using a single photograph, it would correctly re-identify 62% of individuals amongst the faces of all 7.53B humans worldwide, according to our model. Due to both its high entropy ($h = 41.54$) and tail complexity ($\gamma = 0.68$), simple web fingerprints would similarly correctly identify 4B Internet devices with 75% accuracy, a risk feared by security researchers[13,58] but that had never been tested at scale[59].

## Discussion

From statistical disclosure control and computational privacy to browser fingerprinting and machine learning, an extensive literature has been developing techniques to match and identify or re-identify digital traces. Although scholars agree that more auxiliary information and small population size both lead to higher rates of identification, there is no consensus on the nature of the functional form to determine which identification poses a risk when the population of interest is millions or even billions of individuals[17,18,27]. The PYC model provides, for the first time, a principled mathematical model to evaluate how identification techniques will perform at scale. Our findings suggest that two parameters, the entropy and tail complexity, are sufficient to accurately model the identifiability of human data in a wide range of applications. This approach is particularly helpful for novel identification techniques, costly to validate at scale hence often limited to small-scale empirical evidence. Understanding the scalability of identification is essential to evaluate the risks posed by these techniques, including to ensure compliance with modern data protection legislations worldwide.

From a privacy perspective, our model however does not prescribe if a data release is anonymous. For instance, showing that for one identification technique and one set of auxiliary information leads to a low correctness on average is not sufficient to conclude that the risk of re-identification is low. Firstly, a correctness of e.g. 5% still means that some people are identifiable, and techniques exist to single out outlier records in de-identified data[36], synthetic data[60], and machine learning models[61,62]. Secondly, the correctness depends on

**Table 1 | RMSE when extrapolating the correctness $\kappa$ of exact, sparse, and robust matching**

| $\mu$ | Method | c | RMSE | | | | |
|---|---|---|---|---|---|---|---|
| | | | PYC-MB | ENT | EXP | POL | RND |
| | Exact | 216 | **0.290** | 0.370 | 0.315 | 0.484 | 0.347 |
| 0.1% | Sparse | 13 | 0.326 | 0.484 | **0.302** | 0.395 | 0.345 |
| | Robust | 30 | **0.234** | 0.523 | 0.340 | 0.386 | 0.287 |
| | Exact | 290 | **0.153** | 0.364 | 0.287 | 0.355 | 0.305 |
| 0.5% | Sparse | 16 | **0.191** | 0.430 | 0.330 | 0.369 | 0.308 |
| | Robust | 44 | **0.154** | 0.479 | 0.346 | 0.312 | 0.287 |
| | Exact | 316 | **0.122** | 0.357 | 0.262 | 0.330 | 0.282 |
| 1% | Sparse | 16 | **0.104** | 0.422 | 0.253 | 0.393 | 0.287 |
| | Robust | 47 | **0.183** | 0.463 | 0.310 | 0.433 | 0.254 |
| | Exact | 351 | **0.067** | 0.310 | 0.192 | 0.179 | 0.230 |
| 5% | Sparse | 17 | **0.056** | 0.373 | 0.149 | 0.149 | 0.233 |
| | Robust | 58 | **0.134** | 0.411 | 0.224 | 0.361 | 0.216 |
| | Exact | 365 | **0.051** | 0.276 | 0.155 | 0.146 | 0.214 |
| 10% | Sparse | 17 | **0.036** | 0.340 | 0.124 | 0.092 | 0.215 |
| | Robust | 58 | **0.102** | 0.379 | 0.175 | 0.262 | 0.191 |
| | Exact | 400 | **0.013** | 0.128 | 0.037 | 0.028 | 0.213 |
| 100% | Sparse | 18 | **0.013** | 0.152 | 0.051 | 0.055 | 0.202 |
| | Robust | 58 | **0.060** | 0.256 | 0.099 | 0.155 | 0.172 |

We report the results for all selected data collections, grouped by method, for all sampling fractions $\mu$ in 0.1%, 0.5%, 1%, 5%, 10%, 100%. For each data collection, we measure the empirical correctness from $n^{(0)} = 1$ to $n^{(t)} = n'\mu$ records, fit four functional forms, and report the mean RMSE between empirical and estimated correctness of $n$ records. The four functional forms are PYC-MB (Pitman-Yor Correctness functional form, 2 degrees of freedom), ENT (Entropy baseline with no tail complexity, 1 d.o.f), EXP (exponential decay function, 2 d.o.f), POL (polynomial function, 2 d.o.f.). We report results for RND (random) where the value for $\kappa$ is draw uniformly between 0 and $n^{(t)}$. We indicate in bold the method with the lowest error rate. We also report the number of data collections c included from each corpus, left after selecting only data collections for which $0.01 < n^{(t)} < 0.99$. Tables S3-5 show the detailed results per corpus.

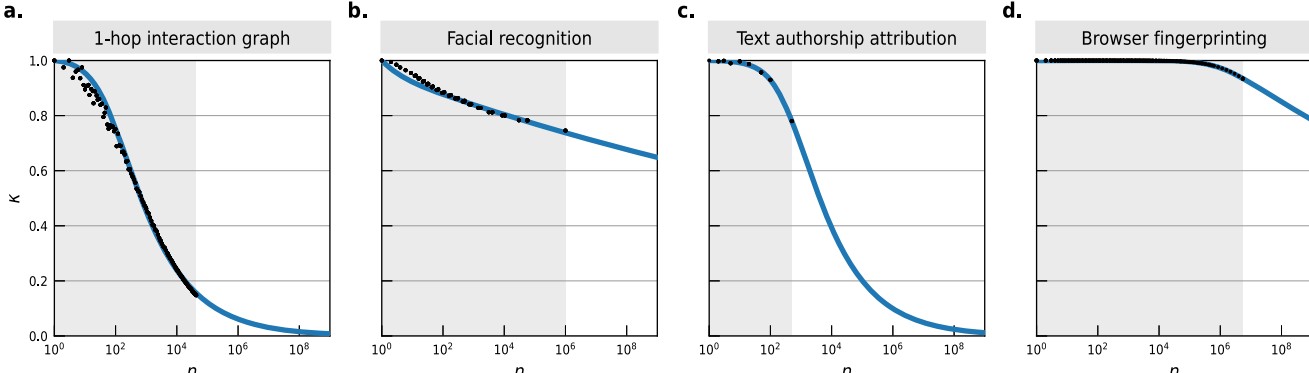

**Fig. 3 | Forecasting the correctness of popular identification techniques with the PYC-MB model.** Each panel shows the empirical correctness $\kappa$ (black dots) in four identification scenarios, along with our prediction $\hat{\kappa}$ (solid blue line) fitted on the empirical $\kappa$ scores. **a** Identification of mobile phone users from their pseudonymized 1-hop social network (IIG-1, $n = 43,000$ phones) by Crețu et al.[25]. **b** Facial recognition using Google FaceNet V8 (FACEREC-2, $n = 1M$ faces) by Kemelmacher-

Shlizerman et al.[56]. **c** Authorship attribution in textual data using Deep Learning (TEXT-1, $n = 500$ authors) by Saedi et al.[29]. **d** Exact matching using simple browser fingerprints (HTTP accept, cookies and JavaScript enabled, timezone, display size, installed fonts, plugins, user agent, video) collected by Panopticlick (WEB-2, $n = 5.5M$ fingerprints)[57].

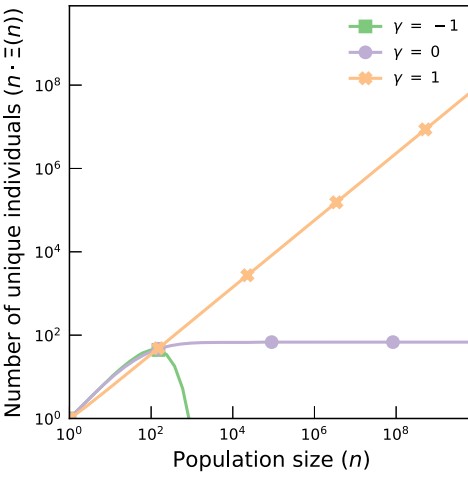

**Fig. 4 | Regimes for the number of unique records.** We report the expected number of population unique records $\mathbb{E}[n \cdot \Xi \mid h, \gamma]$ for $\gamma = -1$ (finite uniform distribution of anonymity sets; green square), $\gamma = 0$ (geometric tail; purple circle), and $\gamma = 1$ (heavy tail; orange cross), for a fixed entropy $h = 10$bits and a population ranging from $n = 1$ to 8B.

the choice of attack model and identification technique used by the attacker and auxiliary information known by them. To evaluate the risk, one would likely need to consider a range of attackers, both strong and weaker ones, and identification techniques[63,64]. However, and conversely, showing under reasonable assumptions that the average correctness is high using our method would likely lead to the conclusion that a risk exists.

From a biometrics and AI harms perspective, we believe that our work can support analysis of robust behavioral identification in particular when deployed in high-risk settings[31] such as in hospitals[32,33], humanitarian aid delivery[34], or border control[32,35]. However, as for the privacy case, a high correctness on average might not be sufficient on its own to analysize the risk posed by the deployment of a AI identification system. For instance, a high correctness of 99% on average still implies that 1% of the population is misidentified, potentially systematically the same group[65]. On the other hand, a low or even medium correctness could mean that a system is not suitable to be used in practice.

While we focus on the correctness, our approach can be readily used to model other privacy and biometrics metrics from the literature such as uniqueness, the fraction of unique individuals in a data collection ($\Xi$)[11,36,66–68] and $k$-anonymity, a popular guarantee to ensure that at least $k$ individuals in a data collection are indistinguishable from one another for any combination of $\phi(x)$[69]. In SI Section S2.1, we derive analytical expressions for the expected uniqueness $\Xi$ of a data collection and the likelihood $V_k$ of $k$-anonymity violations (counting how many records will be part of an anonymity set of fewer than $k$ records amongst a data collection of $n$ records) under a PYP prior. Using the maximum a-posteriori (MAP) estimates of $h$ and $\gamma$, we achieve a low RMSE for both metrics on the 400 exact matching data collections considered (see Table S1).

Above, we note that forecasting low average correctness is not sufficient to conclude that the risk posed by a data release is low. Likewise, forecasting a very high correctness is not enough to prove that a new biometric method will not systematically misidentify specific individuals and minorities. Predicting which records have the least or most chance of being identified is difficult in the setting we consider here. The intrinsic input of our method is the results of a Bernoulli trial, the fact that $m$ attempts at identifying people have been made out of a total $n$. The average thus captures all the information there is. Our work could be extended, e.g. by using individual

correctness for exact matching[36], or by taking into account confidence scores from machine learning models[60,61]. This will, however, require significant mathematical work and grounding into case-specific threat models.

The PYC model can be used, in specific cases, to reason about maximum risks of identification. In the context of exact matching, we can estimate how many records are at maximum correctness, with a standard metric: the fraction of people who will always be identified if matched, the (population) uniqueness $\Xi$. Figure 4 shows that the (absolute) number of individuals uniquely identified continues to grow as the population rises when the distribution of auxiliary information is heavy-tailed ($\gamma > 0$). It asymptotically stabilizes when the distribution is geometric ($\gamma = 0$) and asymptotically decreases to zero when the distribution is finite ($\gamma < 0$). In SI Section S3.4, we show that accurately predicting $\kappa$ implies accurately predicting $\Xi$, suggesting that the 'maximum correctness' can be accurately predicted from a model fitted on correctness measurement scores.

An extensive literature in statistical disclosure control attempts to identify if a match, found in a random sample of disclosed discrete data, will be unique in the complete population, hence modeling the uniqueness $\Xi$ of exact matching. These data-dependent methods include, e.g., extrapolations of the contingency table of the data sample[66,70–77] and generative copula-based methods[36]. These methods are however limited to exact matching, specific types of data (discrete multivariate data with low dimensionality, such as census records) and, crucially, require access to samples of the data—data that is rarely available. Instead, our PYC-MB method can estimate the scalability of identification for exact, sparse, and robust matching purely based on reported correctness measurements or other accuracy metrics such as the uniqueness.

The entropy has long been used as a rule of thumb to evaluate the anonymity and the risk of re-identification in data collections[39,78–82], with a popular saying suggesting that 33 bits of information are sufficient to identify anyone on Earth[57,83–87], 28.2 bits in the USA[83], and 24 bits in Hungary[87]. This approach, however, relies on the simplifying assumption that the underlying data follows a uniform discrete distribution[78,88]. This strong, and often largely incorrect, assumption means that the entropy alone cannot accurately model the correctness. As shown in Table S1 (Supplementary Note S3.1), an entropy-based model (ENT) indeed obtains a RMSE of 0.187, 10.8 times higher on average than the PYC across all 400 surveyed datasets.

Ranging from $\gamma = -1$ for finite uniform distributions to $\gamma = 0$ for geometric tails and $\gamma = 1$ for heavy-tailed ones, the tail complexity has a strong impact on the correctness. Figure 5a shows for instance how at $h = 40$ bits the correctness for the world population can go from 99% when $\gamma = 0$ (geometric tail) to 45% when $\gamma = 1$ (heavy tail). Importantly, distributions with a low tail complexity ($\gamma = 0$, geometric tail) exhibit a critical correctness regime as the entropy grows (Fig. 5b). This means that a small increase of entropy can yield a significant increase of the correctness, from almost 0 to 100% when $\gamma = 0$. This criticality does not occur with a heavy tail: a small increase of entropy only yields a small increase of correctness (Fig. 5b).

Previous studies have also used curve-fitting techniques to extrapolate measurements of privacy metrics from small sample measurements of sparse[17,18] and robust[28,89,90] matching. Exponential decay functions have been used by Achara et al.[17] to extrapolate the unicity of installed smartphone apps and by Sekara et al.[18] for human mobility traces, as well as used in the NIST's Face Recognition Vendor Test (FRVT) to forecast the accuracy of facial recognition technologies[28]. Similarly, polynomial functions have been used by Friedman et al.[89] and Baveja et al.[90] to extrapolate the correctness of facial recognition. Table 1 and Tables S3–7 compare the performance of the proposed functional forms to the PYC-MB method (Supplementary Note S3.2). Across all sampling fractions and types of

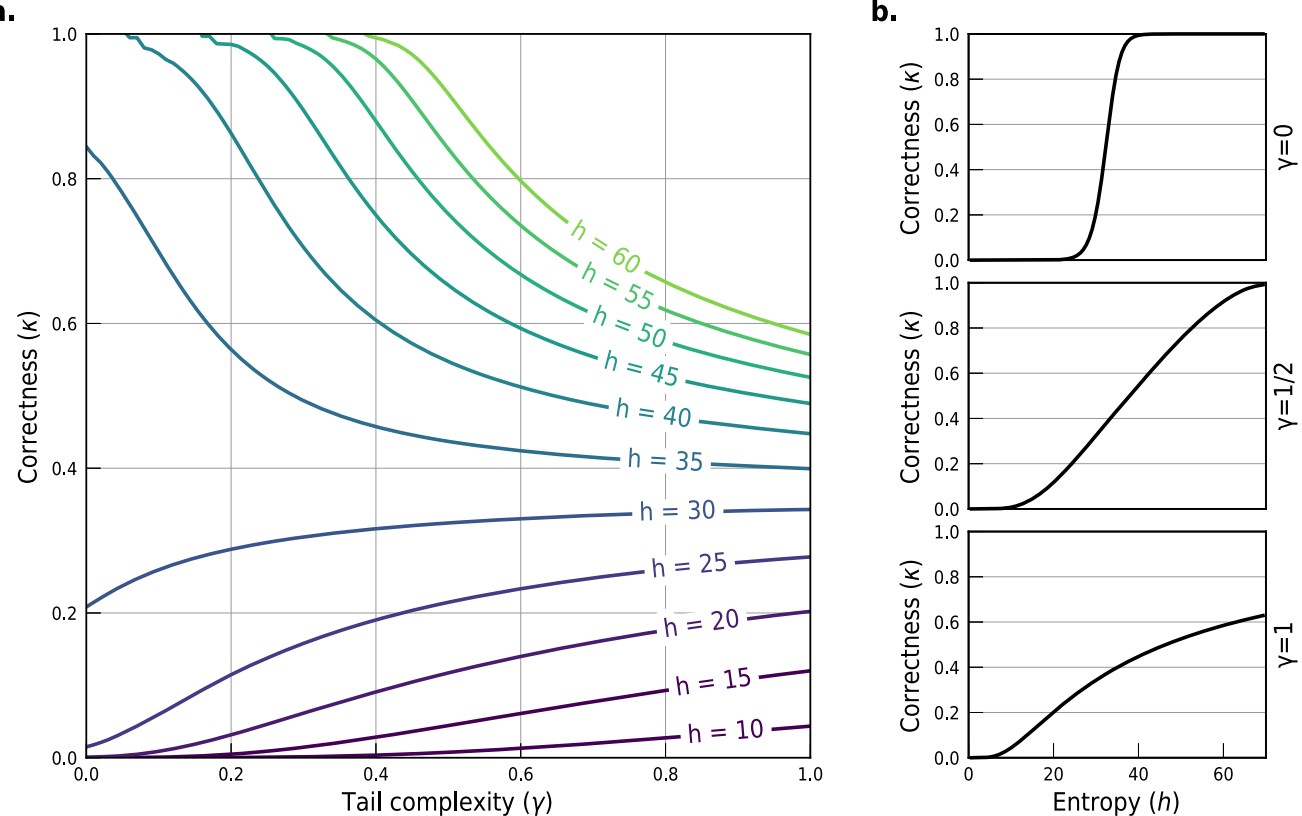

**Fig. 5 | Criticality and regimes of correctness.** We report the expected correctness $\mathbb{E}[\kappa \mid h, \gamma]$ in two scenarios for a fixed world population of $n = 7.53$ billion people. **a** Effect of the tail complexity parameter $\gamma$ on the expected correctness. Each line represents the correctness for a fixed entropy $h$ from 10 to 60 bits, with color indicating the entropy $h$. **b** Effect of the entropy $h$ on the expected correctness. Critical behaviors arise for exponential tails ($\gamma = 0$, top) but not for heavy tails ($\gamma = 0.5$, middle; $\gamma = 1$, bottom).

matching methods but one (0.1% for sparse matching), the PYC outperforms often strongly curve-fitting techniques, despite having the same degrees of freedom. This suggests that the PYC-MB and the underlying PYC model offer a principled and accurate approach to model and estimate the correctness, strongly outperforming both previous functional forms and rules of thumb.

## Methods

We study the expected value of privacy metrics, in a gallery $\boldsymbol{G} = \{x_l\}_{l=1}^{n}$ containing $n$ enrolled records, each drawn i.i.d. from the discrete distribution $X$. We model any privacy metric $\Psi_{\boldsymbol{G}}$ defined as the mean of an individual metric $\psi_{\boldsymbol{G}}$, over all its records:

$$\Psi_{\boldsymbol{G}} = \frac{1}{n}\sum_{i=1}^{n} \psi_{\boldsymbol{G}}(x_i) \qquad (5)$$

Within the gallery $\boldsymbol{G}$, the uniqueness $\Xi$ is the expected fraction of records $x \in \boldsymbol{G}$ with a unique set of auxiliary information $\phi(x)$[36], with $X_\phi$ its marginal distribution. We compute its expression using:

$$\psi_{\boldsymbol{G}}(x_i) = \left[\phi(x_i) \text{ unique in } \{\phi(x_i)\}_{i=1}^{n}\right] \qquad (6)$$

In SI Section 2.1, we derive the following closed-form expression for the uniqueness $\Xi$:

$$\Xi(n) = \sum_{x_\phi \sim X_\phi} p(x_\phi)\left(1 - p(x_\phi)\right)^{n-1} \qquad (7)$$

Similarly, we can derive the correctness $\kappa$, the expected fraction of records $x \in \boldsymbol{G}$ correctly matched from their auxiliary information $x_\phi$[36]

using

$$\psi(x, n) = \mathbb{P}(x_\phi \text{ correctly matched in a sample of } n \text{ records}) \qquad (8)$$

In SI Section 2.1, we derive the following closed-form expression for the correctness $\kappa$:

$$\kappa(n) = \sum_{x_\phi \sim X_\phi} p(x_\phi)\frac{\left[1 - p(x_\phi)\right]^{n}}{n\,p(x_\phi)} \qquad (9)$$

Finally, we can derive $V_k$, the expected fraction of $k$-anonymity violations, using:

$$\psi(x, n) = \mathbb{P}(1 \text{ to } k - 1 \text{ records share } \phi(x) \text{ amongst } n) \qquad (10)$$

In SI Section 2.1, with $I$ the regularized incomplete beta function, we derive the following closed-form expression for $V_k$:

$$V_k(n) = \sum_{x_\phi \sim X_\phi} p(x_\phi)\,I_{1 - p(x_\phi)}(n - k + 1, k - 1) \qquad (11)$$

### Pitman-Yor priors and expected privacy metrics

We denote by $PY(d, \alpha)$ the Pitman-Yor process with a discount parameter $d \in [0, 1]$ and a concentration parameter $\alpha \in [-d, +\infty]$. In SI Section 2.1, we show that, for $X_\phi \sim PY(d, \alpha)$, the expected uniqueness in

a gallery of $n$ records is:

$$\mathbb{E}\left[\Xi(n)\,|\,d,\alpha\right] = \frac{\Gamma(\alpha+1)}{\Gamma(d+\alpha)}\frac{\Gamma(n+d+\alpha-1)}{\Gamma(n+\alpha)} \tag{12}$$

We also show that the expected correctness is:

$$\mathbb{E}\left[\kappa\,|\,d,\alpha\right] = \frac{1}{nd}\left(\frac{\Gamma(1+\alpha)}{\Gamma(d+\alpha)}\frac{\Gamma(n+d+\alpha)}{\Gamma(n+\alpha)} - \alpha\right) \tag{13}$$

Using the generalized hypergeometric function ${}_3F_2$, we finally show that the expected fraction of $k$-anonymity violations is:

$$\mathbb{E}\left[V_k(n)\,|\,d,\alpha\right] = \binom{n-1}{n-k+1}\Gamma(k-d)\Gamma(n-k+1+d+\alpha)$$
$$\frac{{}_3F_2(1,n,n-k+1+d+\alpha;n-k+2,n+1+\alpha,1)}{B(1-d,d+\alpha)\Gamma(n+1+\alpha)} \tag{14}$$

### Reporting summary
Further information on research design is available in the Nature Portfolio Reporting Summary linked to this article.

### Data availability
The data generated in this study have been deposited on OSF at https://osf.io/shnrx/. A complete description of all datasets and accession codes is available in Supplementary Information (SI Section S1).

### Code availability
All simulations were implemented in Python. The source code to reproduce the experiments is available at https://osf.io/shnrx/, along with documentation, tests, and examples.

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

## Acknowledgements

We would like to thank members of the Computational Privacy Group, in particular A.-M. Crețu and T. Lienart, as well as members of the Oxford Internet Institute, in particular D. Sutcliffe, for providing comments on an earlier version of this article. L.R. acknowledges support from the Royal Society Research Grant RG\R2\232035, the John Fell OUP Research Fund, and the UKRI Future Leaders Fellowship [grant MR/Y015711/1]. Y.A.dM. acknowledges funding from the Information Commissioner Office.

## Author contributions

L.R., Y.A.d.M., and J.H. conceptualized the study. L.R. derived models and performed data curation, formal analysis, software development, and managed computing resources. L.R., Y.A.d.M., and J.H. wrote the manuscript.

## Competing interests

All authors declare no competing interest.
