## [Transparent Peer Review file · Nature Communications]

A scaling law to model the effectiveness of identification techniques

Corresponding Author: Dr Luc Rocher

Version 0:

Reviewer comments:

Reviewer #1

(Remarks to the Author)

The manuscript proposes a simple two-parameter model for re-identification risk, focusing primarily on the marketer risk (i.e., average correctness of identification).

Overall, the manuscript offers a useful contribution to the field. Nevertheless, it is limited somewhat by several assumptions that it makes. Two in particular seem consequential: a) assumption that the set of attributes known to the adversary is uniformly distributed (experiments seem to hinge on this assumption, which is quite unreasonable in practice), and b) assumption that the particular sparse/robust approaches for re-identification are fixed and known. Let me elaborate a bit.

1) Assuming uniform distribution of attribute knowledge by attacker: in practice, the set of attributes known to the attacker is not uniform, but based largely on side information actually available online. I recommend also referencing Xia et al. Enabling Realistic Health Data Re-identification Risk Assessment Through Adversarial Modeling, JAMIA 2021, and perhaps using the experiments in this work (especially those that do not assume uniform attribute knowledge prior) as the basis for additional evaluation of the proposed ideas.

2) Knowledge of sparse/robust attacks: it is not clear in the paper which particular attacks were used in the experiments, but it seems that these are assumed to be known to fit $k(n)$. It should certainly be clarified what these were. More significantly, I would like to see experiments which investigate what happens if one attack approach is used to train $k(n)$, while evaluation of efficacy is performed using another attack.

Some additional comments:

3) A somewhat more minor point is that the meaning of n_S is never well explained. I inferred early on that it seems to mean the group size given x_ϕ , although later discussion and descriptions of n_S make me less confident that I am correct. There seems a typo on page 9: $n \leq n_S$ should be $n \geq n_S$, I think.

4) Typo: page 9, "information x_ϕ correctly belongs" -> delete "correctly".

5) X_ϕ on page 6 seems to refer to both the distribution of x_ϕ , the set of possibilities (sum over these in (1); actually, the sum term seems technically imprecise, as $x_\phi \sim X_\phi$ denotes distribution of x_ϕ , but then the sum over these does not make sense), and random variable (as in $P(X_\phi = x_\phi)$). This should all be cleaned up, I think.

6) Datasets on page 7 should include references.

Reviewer #2

(Remarks to the Author)

When modelling a set of records, why do you consider that the Pitman-Yor process is a good model? It's hard to see how two parameters would be enough for most datasets derived from real human complexity - my location trace might depend on where I live, what sort of work I do, whether I'm in the habit of bringing my phone with me all the time, whether I'm currently in lockdown, whether I'm well or sick, whether a neighbouring country changes its rules about visas for people of my

citizenship, etc etc. Similarly, it's hard to see a two-parameter model adequately incorporating all the variety of human health records, let alone DNA. I think the paper would be greatly improved if you defined and explained the Pitman-Yor process more clearly.

Is κ a standard parameter in this field, or is it something you devised for this paper? I understand that it is an average quantity, and I question whether the average, rather than the maximum, is the useful quantity here. Suppose you had a dataset in which one person - Alice - could be immediately and correctly identified with very little auxiliary information. Would it be OK to publish that dataset? If you added a lot of other records that were harder to identify, but did not match the auxiliary information on Alice, thus making the average identifiability confidence (κ) decrease but not making Alice any harder to identify, would that make it better or worse? I think this question of what to measure deserves much more careful thought, and I'd like to see analysis of the maximum identifiability as well as (or instead of) the average.

I don't agree that you've demonstrated (as you say on p.12) that "[identification] effectiveness decreases fast as soon as the population of interest is above 100 to a 1,000 people." _Average_ effectiveness is the only thing you've shown to decrease - it's still quite possible that the number of identifiable individuals continues to grow as the population rises.

I found p.6 hard to follow - I wasn't even sure what the types of the various quantities were. I'm assuming that X_{Φ} and x_{ϕ} are vector quantities, because otherwise it would be hard to model identification of multi-attribute datasets, but I'm not sure exactly how you are generating these from a Pitman-Yor process.

It's good to have evaluated this model on real data, but please tell us more about the datasets you used. Links would help. For example, I wasn't aware that the US census exposed individual records - are they treated in advance in some way to remove unusual individuals? Where did you get this data from? Similarly, where did the web fingerprinting data come from? Do you mean browser fingerprinting (e.g. graphics rendering, plugins, etc, as in the EFF's "how unique is my browser" site at <https://coveryourtracks.eff.org/>), or do you mean fingerprinting an individual's web browsing history, which I would think would have _much_ higher entropy and a much thicker tail? I didn't understand your second example at all - please explain "survey (HDV, MIDUS)."

I wouldn't bother comparing your model with synthetic data, or proudly explaining that it models it particularly well - it's real human data that matters in this paper, and my concern is that it is much more complex than either your Pitman-Yor model or your other synthetic data. What would it mean for your results if that 1.3% of bias really was related to the unusual records that were easiest to identify?

Now that I am on p.10 I see that for the 'robust' matching you are assuming $r \geq 2$ observations. It would have helped a lot to say that you considered only one observation on pp.6--7, because it wasn't clear to me whether you were developing the math for only the 'exact' case, or for all three.

(By p.11 I understand that WEB-2 is browser fingerprinting, but I still don't understand what WEB-1 was.)

You also need to distinguish more carefully between different notions of large datasets. I agree that identifying Alice from within a small set is generally easy. Identifying Alice from within a larger dataset (of the same data about each individual) is clearly harder. Agreed. However, there are other situations you need to distinguish more clearly. For example, identifying whether Alice is present in a 10% sample of the population (in which she may or may not be present) is probably easier than identifying whether she is present in a 20% sample (because she's probably not there, and the larger the sample the higher the likelihood of a mistaken overlap in auxiliary data). Some people have argued that releasing sample fractions, rather than the whole population, makes the data harder to re-identify. Do you agree? (I don't, fwiw, but there's a complicated argument about confidence to be made.)

However, as I've said above, the difficulty of identifying _at least one_ person from a dataset does not obviously rise as the dataset gets larger. A larger dataset may have a larger number of very unusual individuals.

Overall, although I think the work is interesting, I think the writing needs a lot of work.

- Define the problem - what exactly is re-identification? What is the overall population from which we're trying to identify someone? Why should we care about the average rather than the worst case? etc. You probably think you already have this clear in your mind, but you haven't expressed it clearly in your paper.

- Explain the Pitman-Yor process precisely, using math, and explain the types of the inputs and outputs, and how they correspond to the data records you've been informally describing.

- Make a more critical analysis of the alignment between PY simulations and real data. (I really don't care whether they accord with synthetic data and wouldn't bother including it.) As I said already, I'm concerned that the (admittedly small) deviations between simulation and reality may be exactly the records that are easiest to identify. Do you agree? Can you think of a way to test this empirically? If it were true, what would it imply? Would your simulated identifiability results be a lower bound on the identifiability of real data?

- Consider redoing all the experiments measuring maximum, as well as (or instead of) average, identifiability. I'd be surprised if things still look better with larger datasets. And I'd argue that if one person is easily identifiable, the dataset shouldn't be released (without that person's consent).

Vanessa Teague, April 13th 2023.

Version 1:

Reviewer comments:

Reviewer #1

(Remarks to the Author)

I appreciate the revisions to the manuscript made by the authors. I believe they have made the paper easier to understand, and addressed my most substantive comments. In particular, the broader impacts statement section is very helpful in better framing the problem being solved; indeed, I would suggest that it, or a part of it (say, the first paragraph) be moved to the introduction (perhaps the rest can then be integrated into the discussion).

There are some remaining relatively minor issues:

- 1) Page 2, introduction: e.g. -> e.g.,; people However -> period missing.
- 2) Page 3, introduction: "We here propose"  delete "here" (it doesn't serve any purpose).
- 3) Page 5, "Importantly, our approach assumes no specific distribution for the frequencies π_i "; it explicitly assumes PY distribution, so this statement seems inaccurate; I suggest rewording.
- 4) Page 6, "from a flexible and correct model": I would delete "correct", since no model is "correct" in an objective sense, just more or less accurate.
- 5) Figure 1: it is interesting to observe that predictions systematically under-predict empirical correctness. This is not the end of the world, but should be commented on.
- 6) Figure 2: is "s" in the plots supposed to be "mu"? I don't see any reference to s in the text, so it seems like this is a typo.

Reviewer #2

(Remarks to the Author)

I think it would help to make the attacker model clearer in the introduction. To put it another way, I'm not really convinced by the high-risk settings at the bottom of p.3. Until now you've been discussing a model whether the attacker is trying to re-identify data, and wins if correctly identifies some fraction of the population. In the case of bank account authentication, the attacker is trying to spoof someone's voiceprint and fool the identification algorithm. This isn't the same thing at all. For example, you could imagine a society of great actors in which everyone's natural speaking voice was distinct and easily identifiable by the auth system, but in which it was very easy to deliberately synthesize someone else's voice and hence get into their bank account. If I understand your paper rightly, you are investigating on the 'natural/non-deliberate' uniqueness and identifiability of individuals, not any kind of deliberate attack to emulate others' data. (I don't know anything about identifying newborns by their irises – sounds hideous – but I imagine the attack model warrants some careful consideration too.)

Definition of k : Is there a reason for restricting to "exact matching" attacks (which are based only on quasi-identifiers) rather than also including "sparse matching" (which can use other data)? You haven't actually defined quasi-identifiers yet anyway. I've always thought it was a problematic concept, because there is no good distinction between "data that can be used for identification" and "data that cannot be used for identification" – so why not just say that your definition of k applies to sparse matching as well? If I understand rightly, the definition would work fine, but would be 100% whenever the sparse data was a unique match for the auxiliary data. (You discuss this further down, but I think it would make more sense just to include sparse matching from the beginning.)

"Without loss of generality, our model assumes that each record $x \in G$ is drawn i.i.d..." That most definitely is a loss of generality. It's fine for a model to make assumptions, but just say you're making an assumption rather than saying it doesn't reduce generality.

Other than those relatively small issues, I think the paper is greatly improved. I appreciate the additional section about max identifiability, and I thank you for improving the clarity of the math.

Report by Vanessa Teague

RESPONSE TO REVIEWERS

Comments to Reviewer #1 and Reviewer #2

Thanks to your respective comments, we realized our contribution could be made more clear.

In our work, we study the average correctness of identification across three types of attacks. The first two, exact matching and sparse matching, are typical attacks in the statistical disclosure control literature. The third one though, robust matching, includes probabilistic matching but also other techniques such as machine learning behavioral identification techniques (facial recognition, authorship attribution, profiling using interaction graph data, etc.).

Across the three types of attacks, our goal is to model—**given both a matching technique and a specific set of auxiliary information**—how the average correctness of identification scales with the population in which the attacker tries to identify the target. We believe this to be an important line of research as many techniques, in particular ML-based techniques, are actually evaluated on small populations. This makes both the risk they pose in setups where the population of interest is larger unclear but, importantly, also prevents evaluating and comparing the efficiency of techniques across a range of population sizes. Importantly, while from a privacy perspective, high accuracy means there exists a risk of re-identification, high accuracy can be a requirement in cases where the legitimate task is to authenticate someone, e.g. to protect access to a bank account.

From a privacy perspective, we see our work (studying the scalability of average correctness of identification given both a matching technique and a specific set of auxiliary information) as a new tool in the toolbox to evaluate the average correctness of identification across population sizes.

In practice, we envision the tool to be used by privacy researchers and practitioners alongside the broad range of existing frameworks and techniques to evaluate the risk posed by a data release. For instance, a practitioner could use our tool to study impact on the scalability of re-identification attacks of the attacker, by comparing access to various sets of auxiliary information, using the same fuzzy matching technique. Here the sets could be the correct demographic information but also a noisy version of it with approximate admission dates to a hospital, or, as you mention below, probabilistic knowledge of quasi-identifiers. Conversely, the tool can also be used to compare different matching techniques on the same set of auxiliary information, to show e.g. that while method A works well at small scale this other method B is more accurate when the population is larger.

Finally, as mentioned by reviewers #2, we agree that a low average risk (given a set of quasi-identifiers and a method) is not sufficient to conclude that a data release is safe.

We now clarify in the manuscript what our contribution is and how we envision it being combined with existing know-how and techniques to study the privacy risk of data releases.

Reviewer #1 (Remarks to the Author):

The manuscript proposes a simple two-parameter model for re-identification risk, focusing primarily on the marketer risk (i.e., average correctness of identification).

Thank you.

Overall, the manuscript offers a useful contribution to the field. Nevertheless, it is limited somewhat by several assumptions that it makes. Two in particular seem consequential: a) assumption that the set of attributes known to the adversary is uniformly distributed (experiments seem to hinge on this assumption, which is quite unreasonable in practice), and b) assumption that the particular sparse/robust approaches for re-identification are fixed and known. Let me elaborate a bit.

1) Assuming uniform distribution of attribute knowledge by attacker: in practice, the set of attributes known to the attacker is not uniform, but based largely on side information actually available online. I recommend also referencing Xia et al. Enabling Realistic Health Data Re-identification Risk Assessment Through Adversarial Modeling, JAMIA 2021, and perhaps using the experiments in this work (especially those that do not assume uniform attribute knowledge prior) as the basis for additional evaluation of the proposed ideas.

We fully agree with the reviewer that, when evaluating the risk posed by a specific data release, one should not solely consider cases where the attacker knows a fixed set of auxiliary information or them to be known uniformly at random. As discussed above, however, we see our method to be a tool to evaluate the average correctness of identification across population sizes given both a matching technique and a specific set of auxiliary information. As such, it is not meant to evaluate the risk of a data release but rather to be a tool to be combined with expertise and models such as the one proposed by Xia et al. to evaluate the risk posed by a data release in practice. We now clarify this in the manuscript and reference Xia et al.

The reviewer is right that, in our experiments, we pick uniformly at random the auxiliary information (attributes) known to the attacker. And, in the context of sparse matching, in line with the literature, we also assume points to be equally likely to be known to the attacker. The goal of our experiments is, however, not to conclude that a specific dataset is or is not at risk of re-identification (something that would require a in-depth analysis of what might be available to various attackers depending on the context, etc) but rather to **empirically test that—once released—our method will accurately evaluate correctness (the average risk of re-identification) across many kinds of auxiliary information that an attacker could know**. Said differently, our goal is to validate that our extrapolation technique works independently of how the known auxiliary information is distributed (which it does). We also note that, while the notion of quasi-identifiers is fairly well defined in some cases (e.g. health data), it might not directly translate to other cases e.g. facial recognition or authorship attribution.

2) Knowledge of sparse/robust attacks: it is not clear in the paper which particular attacks were used in the experiments, but it seems that these are assumed to be known to fit $k(n)$. It should certainly be clarified what these were. More significantly, I would like to see experiments which investigate what happens if one attack approach is used to train $k(n)$, while evaluation of efficacy is performed using another attack.

When you say “assumed to be known to fit $k(n)$ ”, we assume you mean attacks on which we know our model would work well. Our apologies if this was not clear: we did not select specific attacks. Instead, we considered a broad range of exact, sparse, and robust attacks from the literature (and showed our model to work well on all of them). We have clarified this in the manuscript.

On train vs evaluation/test, our apologies but we are not sure what you mean? Our model allows an analyst, given the correctness of an attack (matching method and auxiliary information) e.g. on small populations of size 100 to 10000 to evaluate what the correctness of the attack is likely to be e.g. on a larger population of 1M. As this is really about extrapolating the correctness of one attack (using a simple two parameters model), we are not sure in which cases one would want to “train” the PYC model on an attack (e.g. facial recognition from CCTV images) and evaluate it on another (e.g. authorship attribution from texts posted on social media)?

Some additional comments:

3) A somewhat more minor point is that the meaning of n_S is never well explained. I inferred early on that it seems to mean the group size given x_ϕ , although later discussion and descriptions of n_S make me less confident that I am correct. There seems a typo on page 9: $n \leq n_S$ should be $n \geq n_S$, I think.

We apologize for the confusion which we have now corrected. In the original draft, n_S meant the size of the dataset available to train the PYC model.

This means that, in this particular case, there was no typo: what we mean is that accurately predicting a range of correctness from $n=1$ to n_S suggests that correctness curves can be extrapolated in general to larger populations.

We have now strongly rewritten the Results section to better introduce the context and notations, including the dataset and sample on which small identification studies are performed, and have edited the main text accordingly.

4) Typo: page 9, "information x_ϕ correctly belongs" -> delete "correctly".

Thank you, we fixed that typo.

5) X_ϕ on page 6 seems to refer to both the distribution of x_ϕ , the set of possibilities (sum over these in (1); actually, the sum term seems technically imprecise, as $x_\phi \sim X_\phi$ denotes distribution of x_ϕ , but then the sum over these does not make sense), and random variable (as in $P(X_\phi = x_\phi)$). This should all be cleaned up, I think.

Thank you. We apologize for this abuse of notation. We now better formalized the indexing and clarified each summation term. We believe this makes the model much easier to read and understand.

6) Datasets on page 7 should include references.

Thank you. We provide a detailed description of each dataset in Supplementary Information (SI section S1, Data description and data availability) with references. We have edited the section 'Empirical validation of the PYC model' on page 7 to reference each dataset and point to the Supplementary Information for further details.

Reviewer #2 (Remarks to the Author):

When modeling a set of records, why do you consider that the Pitman-Yor process is a good model? It's hard to see how two parameters would be enough for most datasets derived from real human complexity - my location trace might depend on where I live, what sort of work I do, whether I'm in the habit of bringing my phone with me all the time, whether I'm currently in lockdown, whether I'm well or sick, whether a neighboring country changes its rules about visas for people of my citizenship, etc etc. Similarly, it's hard to see a two-parameter model adequately incorporating all the variety of human health records, let alone DNA. I think the paper would be greatly improved if you defined and explained the Pitman-Yor process more clearly.

We fully agree with you that models, statistical or otherwise, whether they have few or many parameters, will never fully capture the complexity of real life.

However, to be useful, models "only" have to sufficiently capture the behavior of interest for the task at hand. To quote one of the statisticians' favorite aphorism: "All models are wrong, but some are useful". The normal distribution, e.g., does not capture the genetics of individuals or their eating habits with its two parameters (mean and standard deviation) but is a useful model when studying the distribution of height in a population for example. Similarly, the 'Susceptible, Infectious, or Recovered' (SIR) model is a two-parameter one model to study the spread of infectious diseases. It doesn't encode population genetics, how specific viruses spread, information about the climate yet it proved itself to be a useful (albeit far from perfect) tool for epidemiological research.

Similarly, we see our model to be a useful tool to estimate how the correctness of identification scales with the population size, on average. It does not at all capture the "real complexity of human life" but we show it to accurately model the quantity of interest (average correctness) across a wide range of settings using only two parameters.

Accuracy: the model accurately approximate correctness across a wide range of settings, systematically outperforming both heuristics and rule-of-thumbs with only two parameters (occam's razors).

Interpretability: The model is simple and its two parameters are reasonably easy to interpret and explain: researchers are now often used to reason with information entropy and with power-law distributions. We derived a model with two interpretable parameters, instead of other Pitman-Yor parameterizations used in statistics.

Is κ a standard parameter in this field, or is it something you devised for this paper? I understand that it is an average quantity, and I question whether the average, rather than the maximum, is the useful quantity here. Suppose you had a dataset in which one person - Alice - could be immediately and correctly identified with very little auxiliary information. Would it be OK to publish that dataset? If you added a lot of other records that were harder to identify, but did not match the auxiliary information on Alice, thus making the average identifiability confidence (κ) decrease but not making Alice any harder to identify, would that make it better or worse? I think this question of what to measure deserves much more careful thought, and I'd like to see analysis of the maximum identifiability as well as (or instead of) the average.

Thank you for emphasizing this.

First, the correctness κ is indeed a standard metric in the literature both in privacy/machine learning and statistical disclosure control (it can be seen as a generalization of uniqueness, basically taking into account the fact that I still identify someone correctly e.g. 50% of the time when the equivalence class is 2 and in machine learning where it is often called top-1 or rank-1 accuracy. It indeed measures the **average** risk of identification, given a matching technique and a specific set of auxiliary information.

Second, we fully agree with you that simply reporting a low average correctness, even in a large population, is not sufficient to conclude that the risk posed by a data release is low. For instance, given a fixed set of quasi-identifiers, a 5% correctness still means that some people are identifiable. This can be an issue, especially in cases where one could know beforehand who the identifiable people are (e.g. a specific subpopulation has a high correctness). Quite a lot of interesting work exists here e.g. for membership inference attacks against machine learning models or synthetic data. Similarly, correctness is always dependent on the attacker being considered, meaning both the matching algorithm and

the auxiliary information assumed to be known by the attacker. As pointed out by reviewer #1, one would also need to consider a range of reasonable attackers, both in terms of matching algorithms and the auxiliary information they have available when using our tool. Conversely though, showing under reasonable assumptions that the average correctness is high would lead to the conclusion that a risk exists. We have now edited the manuscript to make this clear.

Third, we agree that the question of maximum identifiability is an important one. In the context of our work (looking at both exact, sparse, and robust matching techniques), however, it is ill-defined and we believe out-of-scope. Our method is indeed a general one to “extrapolate” average correctness given a matching method and a set of auxiliary information for exact, sparse, and robust matching. The “intrinsic input” of our method is thus the results of a Bernoulli distribution: the fact that m attempts at identifying people have been made out of a total n . The average thus captures all the information there is in the input. One could think about ways to extend our results, e.g. using individual correctness for exact matching or taking into account the confidence score of a machine learning model. From this, one could then try to extrapolate some measure of the maximum risk e.g. the correctness for the top 1% of most identifiable people or TPR at low FPR, a popular metric in machine learning.

We however believe this to be out-of-scope as it (a) would require very significant theoretical and empirical work to extend and validate our results, including the theoretical grounding for the work, to something like the top 1% most identifiable people (b) the metric we study, average correctness is a fairly stable one in a statistical sense¹ and is thus a quantity that can be reasonably estimated. On the other hand, “maximum risk” metrics like correctness for the top 1% is intuitively likely to be less stable statistically and therefore harder to estimate. Intuitively, adding 100 people to a dataset of 10000 will not change the average risk by much but could dramatically change the maximum risk.

Possibly more importantly, we also believe that when it comes to re-identification a large part of maximum in “maximum risk” should actually be with respect to auxiliary information and matching technique:

1. Intuitively, in the context of sparse matching, maximum would mean persons that are correctly identified given very few specific points rather than given a set of points how often is the person correctly identified. Indeed, we believe that auxiliary information is rarely a “rock solid” fact (the attacker knows DOB and zip code but is unlikely to know the admission date at a hospital). The randomness over which maximum is computed should thus include (as mentioned by reviewer #1) the uncertainty over the auxiliary information and not “just” maximum given a fixed set of auxiliary information. We would like to thank both reviewers for helping us spell this out.
2. We would like to mention that, increasingly, properly evaluating the risk including maximum risk will require ensuring that one is using the state-of-the-art matching technique. In a recent piece of work, we e.g. showed how a new machine learning based approach could outperform previous matching methods by a large margin, for location data from 31.8% up to 58.7% (Tournier et al., 2022²) and graph data from 0.3% up to 52.4% (see Crețu et al., 2022³).

Finally, we think that future work should not only consider the most-likely identification but also the least-likely identification. As discussed before, there indeed exist a wide range of applications where the goal is to ensure everyone is correctly identified. We believe our method can be a useful tool for e.g. researchers and journalists, who currently lack tools to understand e.g. if new AI-based identification methods are robust enough at scale. For instance, facial recognition has been shown to work well on white faces, on which models have been trained extensively, and much less on subjects with darker skin color. This can have negative consequences for people who are misidentified⁴ and has been flagged as an important question in AI fairness⁵.

In the new version of the manuscript, we have now:

¹ See e.g. Donald W. K. Andrews. “Stability Comparison of Estimators.” *Econometrica* **54**, 5 (1986): 1207–35. <https://doi.org/10.2307/1912329>.

² Arnaud J. Tournier, Yves-Alexandre de Montjoye. Expanding the attack surface: Robust profiling attacks threaten the privacy of sparse behavioral data. *Sci. Adv.* **8**, eabl6464 (2022). <https://www.science.org/doi/full/10.1126/sciadv.abl6464>.

³ Ana-Maria Crețu et al. Interaction data are identifiable even across long periods of time. *Nat Commun* **13**, 313 (2022). <https://www.nature.com/articles/s41467-021-27714-6>.

⁴ Tate Ryan-Mosley. The new lawsuit that shows facial recognition is officially a civil rights issue. MIT Tech Review. 2021. <https://technologyreview.com/2021/04/14/1022676/robert-williams-facial-recognition-lawsuit-aclu-detroit-police/>.

⁵ Joy Buolamwini and Timnit Gebru. “Gender shades: Intersectional accuracy disparities in commercial gender classification.” Conference on fairness, accountability and transparency. PMLR, 2018. <https://proceedings.mlr.press/v81/buolamwini18a/buolamwini18a.pdf>

- Clarified our contribution to better explain what our method is doing (modeling correctness given a matching method and auxiliary information) and what can and cannot be concluded based on its results (high correctness is likely an issue, low correctness is not sufficient).
- Discuss (in Discussion) the extension of the work to worst-case risks.
- Performed a new experiment to show that, in specific cases, we can estimate how many people are at maximum risk of identification (see below). In the context of exact matching, we can estimate how many records are at maximum correctness, with a standard metric: the fraction of people who will always be identified if matched. This quantity is often referred to as the (population) uniqueness.

I don't agree that you've demonstrated (as you say on p.12) that "[identification] effectiveness decreases fast as soon as the population of interest is above 100 to a 1,000 people." _Average_ effectiveness is the only thing you've shown to decrease - it's still quite possible that the number of identifiable individuals continues to grow as the population rises.

We have now edited the manuscript in response to your comment. In line with your comments, we make it clear that what we show is that the average correctness decreases.

You also raise a very interesting point in terms of how, in some cases, the metric of interest would be the number of identifiable individuals. This is actually something the model can, very interestingly in our opinion, help reason about. We have thus now added a new experiment in discussion, a use case for our model, showing that the (absolute) number of individuals uniquely identified continues to grow as the population rises when the distribution of auxiliary information is heavy-tailed ($\gamma > 0$), that it asymptotically stabilizes when the distribution is geometric ($\gamma = 0$), and that it asymptotically decreases to zero when the distribution is finite ($\gamma < 0$). See the figure below for an example for n between 1 and the world population (8B), for $h = 10$ bits and according to the PYC model. We believe these new results to further emphasize the value of our two-parameters model.

I found p.6 hard to follow - I wasn't even sure what the types of the various quantities were. I'm assuming that X_{Φ} and x_{ϕ} are vector quantities, because otherwise it would be hard to model identification of multi-attribute datasets, but I'm not sure exactly how you are generating these from a Pitman-Yor process.

We are sorry for the lack of clarity. The distribution X_{Φ} is a distribution of equivalence classes of auxiliary information. In practice, we model x_{ϕ} as an integer, encoding the index of an equivalence class.

We have now heavily edited this section to improve the notation and terminology and to better introduce the model.

It's good to have evaluated this model on real data, but please tell us more about the datasets you used. Links would help. For example, I wasn't aware that the US census exposed individual records - are they treated in advance in some way to remove unusual individuals? Where did you get this data from? Similarly, where did the web fingerprinting data come from? Do you mean browser fingerprinting (e.g. graphics rendering, plugins, etc, as in the EFF's "how unique is my browser" site at <https://coveryourtracks.eff.org/>), or do you mean fingerprinting an individual's web browsing history, which I would think would have _much_ higher entropy and a much thicker tail? I didn't understand your second example at all - please explain "survey (HDV, MIDUS)."

Thank you, we have rewritten the Results section to better introduce datasets. We also have added links for each dataset and added a pointer to SI Section 1 where each dataset is described in depth.

Regarding the census dataset: we used 1-percent Public Use Microdata Sample (PUMS) data from the US census. It is publicly available but, to the best of our knowledge, has been slightly transformed.

Regarding fingerprints: we used browser fingerprinting data kindly provided to us by EFF from the Panopticlick/CoverYourTracks project itself. We do not use browser history but agree with you that web browsing history would have a higher entropy and higher tail, with many unique history profiles.

Regarding our second example: we use public US (MIDUS) and French (HDV) surveys of sociodemographic characteristics, psychosocial characteristics, and physical health.

I wouldn't bother comparing your model with synthetic data, or proudly explaining that it models it particularly well - it's real human data that matters in this paper, and my concern is that it is much more complex than either your Pitman-Yor model or your other synthetic data. What would it mean for your results if that 1.3% of bias really was related to the unusual records that were easiest to identify?

We agree with you that what matters is whether the model does on real data, which it does.

We, however, believe that the experiments on synthetic data are useful to validate both the suitability of the model and the effectiveness of the fitting procedure across a very broad range of scenarios. Indeed, while a real world dataset (given a set of auxiliary information) yields one correctness curve, synthetic data allows us to generate a very large (infinite) number of curves with specific constraints allowing us to validate both the model and the fitting procedure across a very broad range of scenarios. This is however, as you emphasize, something that complements not replace the validation on real-world data. We now make this in the manuscript and, in particular, removed the sentence emphasizing that it does particularly well on synthetic data which is indeed not relevant.

Regarding model bias: the model is quite simple and general, which we believe to be a strength, so some level of empirical bias and variance ought to be expected. We however believe both of them to be relatively small meaning that the model is a good one for the (average) correctness.

Regarding model bias and outlier: this is a very interesting point but one that unfortunately would require (as discussed above) very significant work to define across the scenarios reasonable notions of maximum risk. To yet try to partially address your concerns, we added an extra experiment where we compare error on average correctness and error on uniqueness (the fraction of records that are unique, at risk of being 'easily' identified due to their uniqueness). We report that small errors when evaluating the correctness (average identification) do not lead to large deviations of uniqueness (worst-case identification).

Now that I am on p.10 I see that for the 'robust' matching you are assuming $r \geq 2$ observations. It would have helped a lot to say that you considered only one observation on pp.6--7, because it wasn't clear to me whether you were developing the math for only the 'exact' case, or for all three.

Our apologies for the misunderstanding. Your comment made us realize we had at all not clearly explained the two steps we followed when validating the model. We also used the same name "r" for two different variables, something we have now corrected by renaming the number of measurements "t".

More specifically, we first validate that the PYC model can correctly fit the "underlying" frequency distribution of auxiliary information (in the exact matching case). This is what Fig 1. b-i on page six shows, rank (r) is (in k -anonymity parlance) the size of the equivalence class and frequency is the empirical probability of a sample to be in an equivalence class (i.e. an 'anonymity set') of size r . Then, Fig 1 a. and the insets in plots b-i show how, after having fitted the PY model on the empirical frequency distribution, we can use it to accurately infer correctness as a function of the population size. Taken together these show that the PYC model is well specified to both reason about correctness and to infer correctness at various population sizes.

However, in a lot of cases, the “underlying” distribution is not available (in the exact matching case) or doesn’t exist (in the sparse and robust matching cases). What is, however, available across all three cases is the value of correctness for at least 2 population sizes (the setting $t \geq 2$, e.g the fact that correctness is 0.95 for a population of 100, 0.90 for 1000 and 0.85 for 5000). In the second part of the paper, we then develop the PYC-MB model (MB stands for measurement-based, values of correctness). The PYC-MB can be derived from PYC which allows a user to extrapolate, given at least two measurements (both consisting of a population size and correctness) how the correctness will vary as a function of the population (correctness curves). Note that at least two observations are necessary because our PYC-MB model has two degrees of freedom (entropy and tail complexity). We then extensively evaluate the ability of our PYC-MB model to infer the correctness at various population sizes from a handful of measurements. The model does extremely well, with an RMSE of only 5.1 p.p. on average when extrapolating the correctness in a population 10x larger.

(By p.11 I understand that WEB-2 is browser fingerprinting, but I still don’t understand what WEB-1 was.)

Our apologies this was indeed not clear.

Both WEB-1 and WEB-2 relate to browser fingerprinting. We provide in Supplementary Information the exact auxiliary information and matching method for each of these examples. In particular, we have:

- WEB-1: exact matching using two attributes (UserAgent, Video), a small subset of browser attributes to test the model
- WEB-2 exact matching using nine attributes (HTTPAccept, CookiesEnabled, JavaScriptEnabled, Timezone, DisplaySize, InstalledFonts, InstalledPlugins, UserAgent, Video), a standard set of auxiliary information used for browser fingerprinting.

We have now edited the manuscript to better describe the datasets and auxiliary information we use in the main text.

You also need to distinguish more carefully between different notions of large datasets. I agree that identifying Alice from within a small set is generally easy. Identifying Alice from within a larger dataset (of the same data about each individual) is clearly harder. Agreed. However, there are other situations you need to distinguish more clearly. For example, identifying whether Alice is present in a 10% sample of the population (in which she may or may not be present) is probably easier than identifying whether she is present in a 20% sample (because she’s probably not there, and the larger the sample the higher the likelihood of a mistaken overlap in auxiliary data). Some people have argued that releasing sample fractions, rather than the whole population, makes the data harder to re-identify. Do you agree? (I don’t, fwiw, but there’s a complicated argument about confidence to be made.)

Thank you for our comment.

First, we agree with you and disagree that releasing sample fractions, because it creates uncertainty, makes re-identification harder. Indeed, we have actually shown in a previous article ‘Estimating the success of re-identifications in incomplete datasets using generative models’,⁶ that an attacker can estimate the likelihood of a particular match to be correct even in heavily sampled dataset. This means that, for the people whose data is being released, sampling does not change the risk of re-identification. Conversely, sampling might impact the utility of the data e.g. when analysis focuses on low probability events such as rare diseases.

This work, however, doesn’t aim to address the question of sampling as a risk mitigation strategy. We here assume membership to be known, meaning that we know that the targets are in the dataset. What we aim to study is how the correctness of identification varies with “intrinsic” population size using only a handful of measurements. This allows us to help answer questions like whether authorship attribution is likely to work in a scenario where the intrinsic population is 1M (e.g. Reddit’s subreddit) or whether a new model for facial recognition tested on e.g. 10,000 faces would still work amongst 1M.

Importantly, the previous work we did, based on copulas, would not be applicable here for two reasons. First, copulas require access to the raw auxiliary information data while this work only requires access to a handful of measurements (PYC-MB) of correctness at a few population sizes. Second, copulas only work for exact matching where the data is

⁶ Rocher, L., Hendrickx, J.M. & de Montjoye, YA. Estimating the success of re-identifications in incomplete datasets using generative models. *Nature Communications* 10, 3069 (2019). <https://doi.org/10.1038/s41467-019-10933-3>

discrete and tabular.

However, as I've said above, the difficulty of identifying _at least one_ person from a dataset does not obviously rise as the dataset gets larger. A larger dataset may have a larger number of very unusual individuals.

We agree with you. On the one hand, for a given individual already in a dataset, increasing the size of the dataset increases the chance of adding another individual with the same auxiliary information. On the other hand, increasing the size of the dataset also adds more individuals with unusual information. This is a property that the Pitman-Yor process models well, with unusual individuals being sampled from the tail of the distribution. Overall, we now show that—with the exception of distributions with finite support—the number of unique individuals does not decrease with the population size (see Discussion).

Overall, although I think the work is interesting, I think the writing needs a lot of work.

- Define the problem - what exactly is re-identification? What is the overall population from which we're trying to identify someone? Why should we care about the average rather than the worst case? etc. You probably think you already have this clear in your mind, but you haven't expressed it clearly in your paper.

Thank you. In all honesty, your questions helped us better understand the general context (and concerns about average correctness) and hopefully better explain our contributions and the limitations of our work including future work.

- Explain the Pitman-Yor process precisely, using math, and explain the types of the inputs and outputs, and how they correspond to the data records you've been informally describing.

We now explain the mathematical model and the use of Pitman-Yor priors in Results.

- Make a more critical analysis of the alignment between PY simulations and real data. (I really don't care whether they accord with synthetic data and wouldn't bother including it.) As I said already, I'm concerned that the (admittedly small) deviations between simulation and reality may be exactly the records that are easiest to identify. Do you agree? Can you think of a way to test this empirically? If it were true, what would it imply? Would your simulated identifiability results be a lower bound on the identifiability of real data?

Thank you for raising this point. As mentioned above, we have now tested this hypothesis and can report that small errors when evaluating the correctness (average identification) do not lead to large deviations of uniqueness (worst-case identification). SI Section S3.4 and Fig. S1 in the new manuscript shows that, when trained on the entire population for exact matching, correctness and uniqueness bias are both small, ranging from -0.019 to 0.121 for the correctness, and from -0.018 to 0.105 for the uniqueness. The larger the bias on the correctness, the larger the bias on the uniqueness, and vice versa.

Regarding your second question on the lower bound: when evaluating the risk of re-identification, the average correctness can be seen as a lower bound on the worst-case risk: there is always at least one individual with a higher correctness than the average. If the correctness is low, some outlier records could still be very easily identified. However, if the average correctness is high, it means that most people have a high individual correctness.

- Consider redoing all the experiments measuring maximum, as well as (or instead of) average, identifiability. I'd be surprised if things still look better with larger datasets. And I'd argue that if one person is easily identifiable, the dataset shouldn't be released (without that person's consent).

Thank you. We hope to have better explained the context of our work and why we believe "maximum identifiability" to be difficult to operationalize in the general context of our work. As we mention above, our main contribution is a statistical method to "extrapolate" average correctness given a matching method and a set of auxiliary information for exact, sparse, and robust matching. In this setting, worst-case identifiability is difficult to define mathematically and we believe that it would warrant a new line of research. The "intrinsic input" of our method being the results of a Bernoulli distribution, the average thus captures all the information there is in the input. That said, we show with a new experiment that our method can also accurately predict the number of individuals uniquely identified (therefore at maximum risk), for exact matching attacks. Our results confirm your hypothesis: when the distribution of auxiliary information is heavy-tailed or geometric, some individuals remain identifiable no matter the size of the dataset.

Regarding whether a dataset is safe to be released or not: We see our work as a tool in the toolbox used by researchers and practitioners, to evaluate the average correctness of identification across population sizes given both a matching technique and a specific set of auxiliary information. As such, it is not meant to evaluate the risk of a data release but rather to be a tool to be combined with existing frameworks and techniques to evaluate the risk posed by a data release. We agree with you that, in general, if someone can be easily identifiable, a dataset should not be considered anonymous, especially for public release.

Vanessa Teague, April 13th 2023.

REVIEWERS' COMMENTS

Reviewer #1 (Remarks to the Author):

I appreciate the revisions to the manuscript made by the authors. I believe they have made the paper easier to understand, and addressed my most substantive comments. In particular, the broader impacts statement section is very helpful in better framing the problem being solved; indeed, I would suggest that it, or a part of it (say, the first paragraph) be moved to the introduction (perhaps the rest can then be integrated into the discussion).

Thank you for the kind words. We are glad to hear that the new manuscript addresses most of your comments.

We have now added a paragraph in the introduction, summarising the Broader Impact Statement.

There are some remaining relatively minor issues:

- 1) Page 2, introduction: e.g. -> e.g.,; people However -> period missing.
- 2) Page 3, introduction: "We here propose"  delete "here" (it doesn't serve any purpose).
- 3) Page 5, "Importantly, our approach assumes no specific distribution for the frequencies π "; it explicitly assumes PY distribution, so this statement seems inaccurate; I suggest rewording.
- 4) Page 6, "from a flexible and correct model": I would delete "correct", since no model is "correct" in an objective sense, just more or less accurate.

Thank you. We have fixed these four issues in the manuscript.

- 5) Figure 1: it is interesting to observe that predictions systematically under-predict empirical correctness. This is not the end of the world, but should be commented on.

Thank you for raising this point. We are now discussing the potential slight under-prediction in Results. In short, we hypothesise this statistical bias to arise from the choice of prior during PYP model validation. More specifically, in line with the literature, we used an informative prior that weighs less on strongly heavy-tailed distributions.

Importantly though, we also note that the choice of prior is only relevant for section "Model specification of PYC on empirical frequencies" (validating the PYC model) and not when using the functional form.

- 6) Figure 2: is "s" in the plots supposed to be " μ "? I don't see any reference to s in the text, so it seems like this is a typo.

Thank you for spotting this typo. We have corrected it.

Reviewer #2 (Remarks to the Author):

I think it would help to make the attacker model clearer in the introduction. To put it another way, I'm not really convinced by the high- risk settings at the bottom of p.3. Until now you've been discussing a model whether the attacker is trying to re-identify data, and wins if correctly identifies some fraction of the population. In the case of bank account authentication, the attacker is trying to spoof someone's voiceprint and fool the identification algorithm. This isn't the same thing at all. For example, you could imagine a society of great actors in which everyone's natural speaking voice was distinct and easily identifiable by the auth system, but in which it was very easy to deliberately synthesize someone else's voice and hence get into their bank account. If I understand your paper rightly, you are investigating on the 'natural/non-deliberate' uniqueness and identifiability of individuals, not any kind of deliberate attack to emulate others' data. (I don't know anything about identifying newborns by their irises – sounds hideous – but I imagine the attack model warrants some careful consideration too.)

Thank you.

We have now added a new paragraph in the introduction summarising the broader impact statement and threat model.

You are correct that our model focuses on “natural” uniqueness and the “adversarial” bank account example might be confusing. We have removed it and clarified that we do not consider the adversarial setting.

Definition of k : Is there a reason for restricting to “exact matching” attacks (which are based only on quasi-identifiers) rather than also including “sparse matching” (which can use other data)? You haven't actually defined quasi-identifiers yet anyway. I've always thought it was a problematic concept, because there is no good distinction between “data that can be used for identification” and “data that cannot be used for identification” – so why not just say that your definition of k applies to sparse matching as well? If I understand rightly, the definition would work fine, but would be 100% whenever the sparse data was a unique match for the auxiliary data. (You discuss this further down, but I think it would make more sense just to include sparse matching from the beginning.)

Thank you.

We do share your concerns about the idea that an attacker would, somehow, know perfectly some piece of information and never others. This is why we (probably like you) prefer the notion of unicity (sparse) to uniqueness (exact).

However and unfortunately we could not derive a closed form expression for sparse matching. Unicity-based scores indeed measure an average accuracy of identification across all potential sets of p points known by an attacker (all equally likely to be selected in this threat model). This means that the joint distribution of p points is not equivalent when selecting different sets of p points. We tried but could not come up with either a closed form expression or simple distribution by integrating over all potential sets of p points. We thus decided to focus on exact matching first, derive the theoretical results, validate them, and then extend them to sparse and robust matching using the functional form and empirically validate them.

We have clarified this in the text.

“Without loss of generality, our model assumes that each record $x \in G$ is drawn i.i.d...” That most definitely is a loss of generality. It's fine for a model to make assumptions, but just say you're making an

assumption rather than saying it doesn't reduce generality.

Thank you, you are correct and we have removed the "Without loss of generality"..

Other than those relatively small issues, I think the paper is greatly improved. I appreciate the additional section about max identifiability, and I thank you for improving the clarity of the math.

Thank you.

We would like to sincerely thank you for helping us explain the work and implications of the work better across the review process.

Report by Vanessa Teague